# Guest-host doped strategy for constructing ultralong-lifetime near-infrared organic phosphorescence materials for bioimaging

Fuming Xiao[1,4], Heqi Gao[2,4], Yunxiang Lei[1✉], Wenbo Dai[3], Miaochang Liu[1], Xiaoyan Zheng [3], Zhengxu Cai [3], Xiaobo Huang[1✉], Huayue Wu[1] & Dan Ding [2✉]

Organic near-infrared room temperature phosphorescence materials have unparalleled advantages in bioimaging due to their excellent penetrability. However, limited by the energy gap law, the near-infrared phosphorescence materials (>650 nm) are very rare, moreover, the phosphorescence lifetimes of these materials are very short. In this work, we have obtained organic room temperature phosphorescence materials with long wavelengths (600/657–681/732 nm) and long lifetimes (102–324 ms) for the first time through the guest-host doped strategy. The guest molecule has sufficient conjugation to reduce the lowest triplet energy level and the host assists the guest in exciton transfer and inhibits the non-radiative transition of guest excitons. These materials exhibit good tissue penetration in bioimaging. Thanks to the characteristic of long lifetime and long wavelength emissive phosphorescence materials, the tumor imaging in living mice with a signal to background ratio value as high as 43 is successfully realized. This work provides a practical solution for the construction of organic phosphorescence materials with both long wavelengths and long lifetimes.

[1] School of Chemistry and Materials Engineering, Wenzhou University, Wenzhou 325035, P. R. China. [2] State Key Laboratory of Medicinal Chemical Biology, Key Laboratory of Bioactive Materials, Ministry of Education, and College of Life Sciences, Nankai University, Tianjin 300071, China. [3] School of Materials Science & Engineering, Beijing Institute of Technology, Beijing 10081, P. R. China. [4]These authors contributed equally: Fuming Xiao, Heqi Gao. ✉email: yunxianglei@wzu.edu.cn; xiaobhuang@wzu.edu.cn; dingd@nankai.edu.cn

Room temperature phosphorescence (RTP) of organic materials with persistent emissions can mitigate the interference from environmental self-luminescence. Moreover, the organic matter often has advantages of low toxicity and good biocompatibility[1–10]. Therefore, constructing organic RTP materials would benefit tissue imaging, tumor diagnosis, and drug tracking[10–15]. To date, most phosphorescent materials have poor biological tissue permeability because the wavelengths of their emission spectra are short (less than 580 nm)[9,16–26]. This only means phosphors provide good imaging in shallow regions of an organism. Constructing near-infrared phosphorescent materials has gained some urgency. Although some red RTP materials such as boron fluoride, carbazole, and naphthalene diimides have been developed, those with wavelengths exceeding 650 nm are rare[27–32]. Moreover, presumably limited by the energy gap law, phosphorescence lifetimes of these red RTP materials are very short and not conducive for bioimaging (Fig. 1a and Supplementary Fig. 1).

Establishing ultralong-lifetime RTP materials with long-wavelength emission inevitably means lowering the lowest triplet ($T_1$) level of the material. However, lower $T_1$ levels bring two major obstacles to phosphorescence. One of the obstacles is that lower $T_1$ increases the band gap between $S_1$ and $T_1$ ($\Delta E_{ST}$), which is not conducive to intersystem crossing (ISC) of excitons. The other obstacle is that a lower $T_1$ level easily causes excitons to be consumed non-radiatively, resulting in a significant reduction in lifetime and intensity of phosphorescence (Fig. 1b). Therefore, assuming materials already have low $T_1$, improving the ISC capability of excitons and suppressing the non-radiative transitions of excitons are key in achieving ultralong-lifetime near-infrared/NIR phosphorescence. Recently, guest–host materials have gradually attracted more attention[33–40], because the host molecules can inhibit the non-radiative transition of the guest energy in the guest–host system[41–46]. Additionally, several research results have shown that there is a synergy of energy between host and guest molecules, which can assist the guest molecules to transfer the excited state energy effectively[38–42]. Therefore, the guest–host system provides a new strategy for the construction of organic RTP materials with long-wavelength emissions and long lifetimes.

With that in mind, we tried to construct ultralong-lifetime near-infrared RTP materials employing a guest–host doped strategy. The pyrene derivatives with high conjugation are regarded as guests. Their high conjugation can reduce the $T_1$ levels of the molecules, thereby ensuring that the resultant materials undergo long-wavelength phosphorescence. The benzophenone (**BPO**) compound is chosen as the host matrix, which can act as two roles. One role is associated with assisting transfers of guest excitons; the other role is associated with restricting the motion of the guest molecules, thereby suppressing the non-radiative transition of guest excitons (Fig. 1c). The results show that the designed guest–host materials have strong red afterglow visible to the naked eye. By increasing the degree of conjugation of the guest molecules, phosphorescence wavelengths of the guest–host materials are red-shifted from 600/657 nm to 681/732 nm. More importantly, our newly developed guest–host materials have long phosphorescence lifetimes of 102–324 ms. To the best of our knowledge, this is the first RTP material with both long-wavelength (>700 nm) and long lifetime (>100 ms) simultaneously. Comparative experiments of the molten state and the crystalline state prove that the host matrix restricts the motion of guest molecules is a necessary condition for the doped system to have phosphorescence emission. Molecular dynamics (MD) simulations further show that between host and guest there is a strong interaction that suppresses non-radiative transitions of guest excitons. Moreover, experimental results also confirm that host molecules exhibit synergies with guest molecules in excited states. As a proof-of-concept, the materials were used for precise mapping of lymph nodes and labeling of armpit tumors with high signal-to-background ratios (SBR) of 55 and 43, respectively. The long wavelengths help to reduce scattering from tissue and long lifetimes further mitigate the interference from autofluorescence in bioimaging. Thus, phosphorescent materials with both kinds of properties can provide more unambiguous imaging of tumors.

## Results

### Synthesis and photophysical properties
The guest molecules are based on the pyrene unit, to which adding anisole or *N,N*-

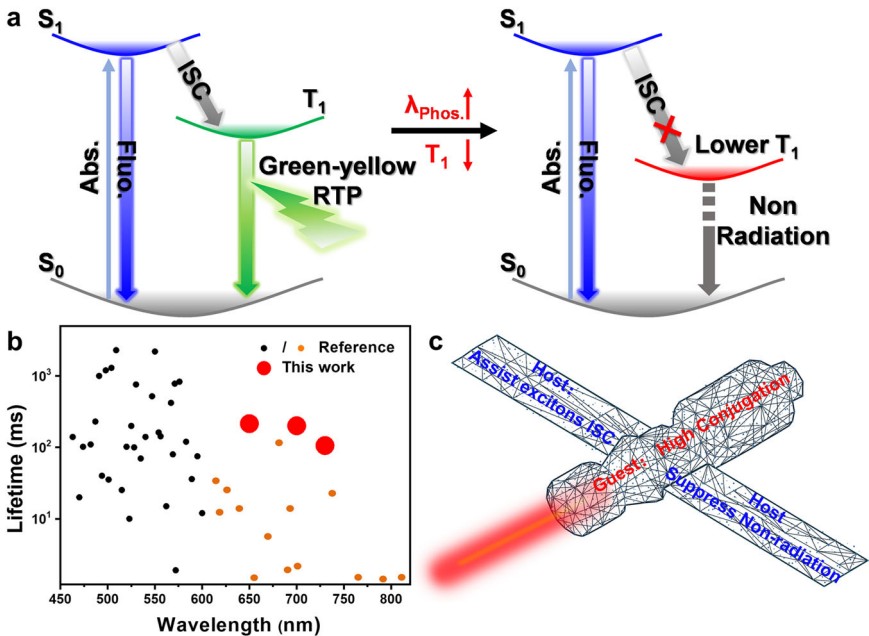

**Fig. 1 Design concept of the guest/host system. a** Problems in constructing ultralong near-infrared RTP materials **b** Phosphorescence performance distribution of RTP materials. **c** Strategy used in this work for constructing ultralong near-infrared RTP materials.

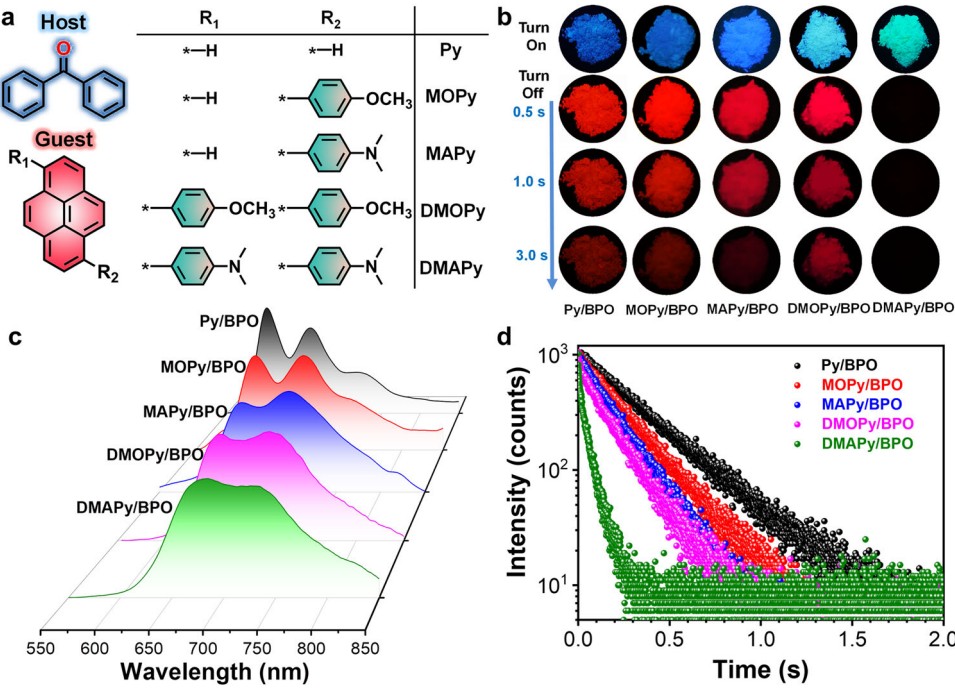

**Fig. 2 Photoluminescence properties of the guest–host system. a** Molecular structures of the guest and host molecules. **b** Fluorescence (top) and phosphorescence (down) images of the guest–host materials. **c** Phosphorescence spectra of host–guest materials. Excitation wavelength: 380 nm; Delayed time: 1 ms. **d** Phosphorescence decay curves of guest–host materials. Excitation wavelength: 380 nm.

dimethylaniline groups on one or both sides through the Suzuki reaction increases the molecular conjugation (Fig. 2a and Supplementary Fig. 2). The five guests show good solubility in chloroform, tetrahydrofuran, and dimethyl sulfoxide. The molecular structures and purities of the target compounds were confirmed by NMR spectroscopy, single-crystal X-ray diffraction, and high-performance liquid chromatography (Supplementary Figs. 3, 4). As guest conjugation increases, the maximum absorption peaks are red-shifted from 343 to 378 nm (Supplementary Fig. 5a), and the corresponding fluorescence peaks are also red-shifted from 381 to 467 nm (Supplementary Fig. 5b). The host **BPO** was purchased directly from a commercial supplier and used without further processing. **BPO** has a low melting point (48 °C) and stable subcooling states and thus the guest molecules can be dispersed in the host using the melt-casting method (The detailed process is in the supplementary information). Because the concentration of the guest molecules is very important in determining the RTP properties of the guest–host materials, we first prepared a series of **Py/BPO** guest–host materials with different guest–host molar ratios (1:50–1:50000) to optimize the luminescence performance. The phosphorescence quantum yields of 1:50, 1:100, 1:500, 1:1000, 1:5000, 1:10000, and 1:50000 are 3.3, 5.5, 7.8, 9.2, 6.8, 5.1, and 3.6%, respectively. The delayed emission spectra further shows that doped material with the guest–host molar ratio of 1:1000 has the strongest phosphorescence intensity (Supplementary Fig. 7), which is in accordance with our previous work[39–42]. Four other guest–host materials (**MOPy/BPO**, **MAPy/ BPO**, **DMOPy/BPO**, and **DMAPy/BPO**, Fig. 2a) with a guest–host molar ratio of 1:1000 were prepared, and the luminescence characteristics of the guest–host systems were systematically investigated.

Five guest–host materials show blue to cyan fluorescence under the excitation source (Fig. 2b) and the maximum emission peaks of the guest–host materials are red-shifted from 416 to 483 nm as guest conjugation increases (Table 1 and Supplementary Fig. 8). Importantly, after finishing the irradiation, with the exception of

**DMAPy/BPO**, the other four guest–host materials have visible to the naked eye a deep red afterglow for ~3 s that reveal the RTP properties (Fig. 2b). Delayed spectra further show that the guest–host materials have two phosphorescence peaks, which are fine structures that arise from energy level vibrations. Similarly, with increasing guest conjugation, the phosphorescence peaks of the guest–host materials are red-shifted from 657 to 732 nm or 600 to 681 nm (Table 1 and Fig. 2c). That is, the guest–host materials produce deep-red or even near-infrared phosphorescence emissions and belong to a group that produces phosphorescence with the longest wavelengths to date. The Commission Internationale de l'Eclairage coordinates further indicate that the guest–host materials have a very deep phosphorescent color (0.63, 0.35; 0.64, 0.34; 0.65, 0.33; 0.69, 0.30; 0.70, and 0.29) (Supplementary Fig. 9). Unlike most red RTP materials which have short phosphorescence lifetimes, the phosphorescence lifetimes of this guest–host system are 102–324 ms (Table 1 and Fig. 2d). Moreover, the guest–host materials have satisfactory luminous intensities, the phosphorescence quantum efficiency being in the range 4.2–9.2% (Table 1). The above results fully prove that, with the guest–host doped strategy, we have developed a group of ultralong-lifetime near-infrared RTP materials. As we know that the stability of materials will greatly affect practical applications. To verify the stability of the doped materials, **Py/ BPO** and **MAPy/BPO** were chosen to soak in water for 24 h, and two materials still have a strong red afterglow (Supplementary Fig. 12). And the **Py/BPO** and **MAPy/BPO** can still maintain obvious red afterglow after being continuously irradiated for 24 h (365 nm, 40 µW/cm²) (Supplementary Fig. 12). In addition, even if the doped materials are left in the ambient environment for one month, the phosphorescence activities are basically unchanged. Therefore, the RTP properties of these materials are very stable to water, light, and air.

Generally, the triplet excitons are unstable and easily assimilated by the motion of molecules, leading to the quenching of phosphorescence. However, for the guest–host system, the host

**Table 1 Photophysical data of the guest-host materials.**

| Sample | Fluo. | | | Phos. | | |
|---|---|---|---|---|---|---|
| | $\lambda_{em}$ (nm) | $\Phi_F$ (%) | $\tau$ (ns) | $\lambda_{em}$ (nm) | $\Phi_P$ (%) | $\tau$ (ms) |
| Py/BPO | 415 | 14.2 | 1.43 | 600[a], 657[b] | 9.2 | 327[a], 324[b] |
| MOPy/BPO | 424 | 13.4 | 2.03 | 623[a], 680[b] | 8.0 | 215[a], 210[b] |
| MAPy/BPO | 440 | 15.2 | 1.98 | 643[a], 697[b] | 6.3 | 201[a], 198[b] |
| DMOPy/BPO | 471 | 12.3 | 2.19 | 657[a], 713[b] | 5.4 | 180[a], 175[b] |
| DMAPy/BPO | 483 | 16.1 | 2.32 | 681[a], 732[b] | 4.2 | 106[a], 102[b] |

Ex. of Fluo.: 360 nm; Ex. of Phos.: 380 nm; Delayed time: 1 ms.
[a]The short phosphorescence peak.
[b]The long phosphorescence peak.

matrix can provide a rigid environment to restrict the motion of guest molecules, thereby ensuring the guests emit phosphorescence[42,43]. We first register the phosphorescence of the solution and solid guests at low temperature (77 K) to verify that the phosphorescence from the guest–host system is emitted by the guest molecules. The spectra show that the guests in the solution state have two fine peaks at 77 K, and the emission peaks are also red-shifted from 596/665 nm to 652/725 nm (Supplementary Fig. 13). The spectral data were almost completely consistent with the phosphorescence wavelengths of the corresponding doped materials. The results confirm that the phosphorescence in the guest–host system is emitted by the guest molecules. Taking advantage of the low melting point of the host, the influence of the host morphology on the phosphorescence performance of the guest–host system was analyzed. The **Py/BPO** molten state at room temperature (subcooling state) show only fluorescence but no phosphorescence (Fig. 3a, b). However, when the guest–host material begins to crystallize, the material produces a bright red phosphorescence. This clearly proves that the host matrix restricts the guest molecular motion and is a necessary factor in determining the RTP properties of the guest–host system.

**Molecular dynamics simulations**. The local microenvironment of the molecules such as the molecular configuration, intermolecular distance, and intermolecular interaction plays an important role in determining the photophysical phenomena of materials. However, obtaining the co-crystal of host–guest is difficult because of the trace amounts of guest molecules (<0.1%) in the entire materials. Moreover, the traditional characterization methods such as X-ray diffractometry, scanning electron microscopy, and transmission electron microscopy are difficult to apply in investigating the molecular conformation of the guests in the host matrix in detail. Therefore, we simulated the molecular conformations of **Py** molecules in the **BPO** matrix using molecular dynamics/MD simulations[40]. The initial **Py/BPO** model was based on the **BPO** crystal. A **BPO** molecule is removed from the **BPO** crystal (Fig. 3c) and a **Py** molecule is inserted into the vacancy. This **Py/BPO** model has a 1:191 molar ratio of **Py** to **BPO**. Starting from the initial Py/BPO configuration, we performed production MD simulations for 10-ns to relax the whole guest–host system using the GROMACS software package (version 5.1.5, details in Supplementary Information). Compared with the conformations of **BPO** molecules in a single crystal, the corresponding conformations of the **BPO** molecules adjacent to the guest **Py** molecule in the **Py/BPO** guest–host system are slightly different because twisting increases the angles slightly after doping (Supplementary Fig. 14). This is because the spatial volume of the **Py** molecule is larger than that of the **BPO** molecule. However, because the number of guest/**Py** in the guest–host system is very small, the impact on the overall

arrangement of the host matrix is minimal. Therefore, the stacking of **BPO** molecules in the simulated guest–host system is almost the same as for a single crystal (Supplementary Fig. 15, Supplementary Fig. 16). The XRD results also confirm little change in the arrangement of the **BPO** host before and after doping with guest molecules (Supplementary Fig. 17). Therefore, we have a microenvironment model of the guest–host system that is reasonable and reliable for our MD simulations.

With this **Py/BPO** model, we first analyzed the relative spatial positions of the Py molecule in the **BPO** matrix (Fig. 3d). The distances between the guest and host molecules in the six directions (up, down, front, back, left, and right) range from 2.3 to 3.1 Å. That is, the guest molecules are in a relatively dense matrix environment, which can effectively inhibit their motion. More importantly, although these distances are relatively similar, the host has a twisted molecular conformation. Therefore, there is no π---π interaction between host and guest, and hence it is not conducive for luminescence. In contrast, between the host and **Py** molecules, multiple C-H---π interactions are evident over short distances (2.3–3.2 Å, red line) (Fig. 3e), and the average distance between a **Py** molecule and surrounding host molecules is only 2.77 Å. In addition, the C-H---O interactions between Py and host molecules are also evident over short distances (2.5 and 2.6 Å, blue line) (Fig. 3e). The above analysis shows that the **Py**-doped host matrix provides a relatively close and strongly interactive environment for guest molecules that effectively restricts the non-radiative decay channels.

**RTP mechanism study**. The rigid environment provided by the matrix is necessary for the guest–host system to display RTP characteristics. However, is this the only role the host molecules play? We chose separately as host sulfonyldibenzene (**SOB**), sulfinyldibenzene (**SIB**), and diphenylphosphine oxide (**PPO**), which also have good crystallinity and a similar structure to **BPO** (Fig. 4a), and **MAPy** as guest. The three guest–host materials (**MAPy/SOB**, **MAPy/SIB**, and **MAPy/PPO**) were prepared with a guest–host molar ratio of 1:1000. Unfortunately, although these materials have strong cyan fluorescence under a UV lamp, no red afterglow appears once the UV source is removed (Fig. 4b). The fluorescence spectra of the three guest–host materials show wavelengths centered around 430 nm (Supplementary Fig. 18a), with quantum yields as high as 63, 71, and 76% in the order given above. Such high luminous intensities indicate that the host indeed inhibits the motion of the guest molecules. However, the delayed spectra suggest that the guest–host materials have almost no phosphorescence emission (Supplementary Fig. 18b). Note that, although the phosphorescence from **MAPy/SOB** and **MAPy/SIB** are very weak, there is an emission peak near 670 nm that once again demonstrates that the phosphorescence in the guest–host material is emitted by the guest molecules. The above comparative experiments show that the rigid restrictive

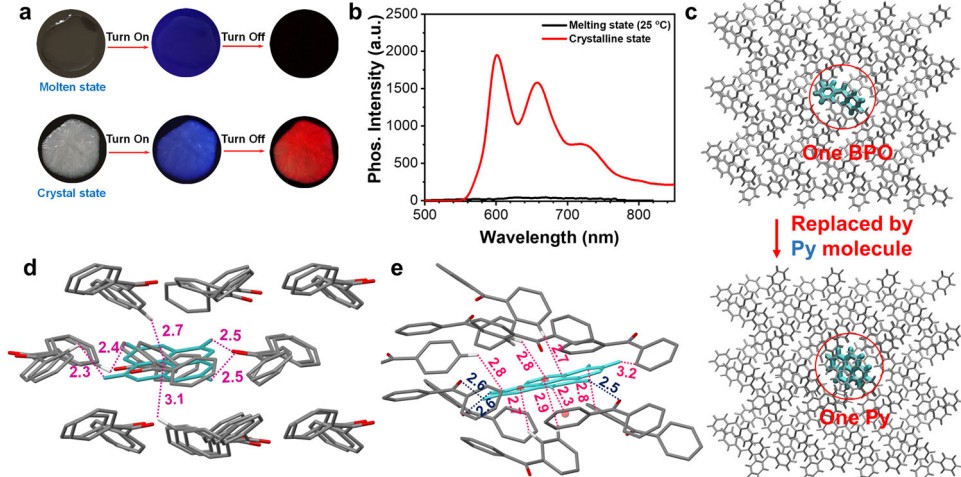

**Fig. 3 Non-radiative suppression of guest excitons by host matrix. a** Photographs of **Py/BPO** in different states. **b** Phosphorescence spectra of **Py/BPO**. Excitation wavelength: 380 nm; Delayed time: 1 ms. **c** Model setup of **Py/BPO** guest–host system. **d** Spatial distances between the **Py** molecule and the surrounding **BPO** molecules. **e** Interaction distances of C-H---π or C-H---O interactions between **Py** molecule and surrounding **BPO** molecules. The distances between each phenyl ring center of **Py** molecule and the hydrogen atom of the **BPO** molecules are marked by a red line. The corresponding distances between the oxygen atom of **BPO** and the hydrogen atoms of **Py** are marked by a blue line.

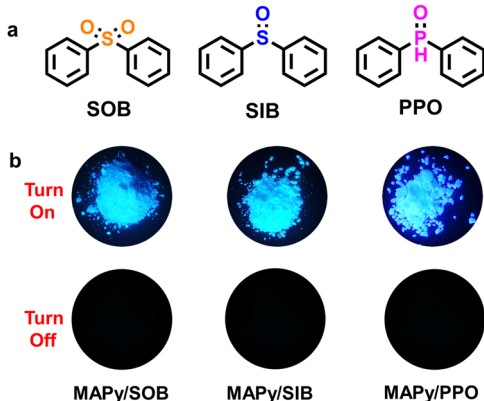

**Fig. 4 Luminescence properties of guest molecules in other hosts. a** Molecular structure of the reference hosts. **b** Luminescent images of the reference guest–host materials.

environment provided by the host is a necessary but not a sufficient factor, for the guest–host materials to display RTP properties.

Energy transfers between host and guest molecules have gradually been revealed to play a vital role in phosphorescence activity. Among them, Förster resonance energy transfer (FRET) is considered a viable explanation why some guest–host materials have RTP properties[23,38]. To verify whether there is a FRET between host and guest, the absorption and excitation spectra of host **BPO** and guest MAPy were investigated. The absorption and excitation wavelengths of host **BPO** (Fig. 5a) only reach 418 nm, whereas the absorption and excitation wavelengths of the guest **MAPy** reach 465 nm. Therefore, we investigated the phosphorescence emission of **MAPy/BPO** material at different excitation wavelengths. The results show that even if the excitation wavelength is extended to 440 nm, the **MAPy/BPO** powder maintains a strong phosphorescence emission (Fig. 5b) and has a red afterglow visible to the naked eye after removing the 420-nm UV source (Fig. 5c). These results clearly demonstrate that the phosphorescence of **MAPy/BPO** does not come from the energy

absorbed by the host matrix, but from the energy absorbed by the guest molecules. Therefore, a FRET between host and guest is ruled out.

In our previous work, we found that the host could assist the excitons of the guests in energy transfer[40–42]. We, therefore, recorded the excitation spectra (fluorescence emission/420 nm) of guest **MAPy** in common solvents (toluene, THF, and N,N-dimethylformamide/DMF) and host **BPO** (molten state and crystal state). The results (Fig. 5d) show that the maximum excitation wavelength in **MAPy** emissions in these solvents is 346 nm, whereas the maximum excitation wavelengths in host emissions in the molten and crystalline states are red-shifted to 392 and 387 nm, respectively. The excitation spectra of the phosphorescence emission (660 nm) also show that the maximum excitation wavelengths in emissions from **MAPy** in these solvents are significantly longer than that of the crystalline host (Fig. 5e). Hence, we conclude that the host not only acts as the rigid matrix, but also changes the energy transfer process for the guest in the excited state. Furthermore, the phosphorescence lifetimes of guest molecules at 77 K are only between 12–23 ms (Fig. 5f), which are much shorter than that of the host matrix. This also shows that the host matrix prolongs the ISC process of the guest excitons.

From the above experimental results and our previous work[40–42], we conjecture that the $T_1$ of the host is the bridge between $S_1$ and $T_1$ of the guest (Fig. 5g), and hence the excited energy from the guest is transferred from $S_1$ to $T_1$ of the guest via the $T_1$ path of the host. To verify this mechanism, density functional theory calculations were performed to obtain the singlet and triplet energy levels of the guests and host (Calculated the energy level of the host in the crystalline state). The energy range for the $S_1$ and $T_1$ states of the five guest molecules are 3.01–3.49 eV and 1.84–2.15 eV, respectively (Fig. 5h). The $\Delta E_{ST}$ of the guest molecules are in the range of 1.17–1.34 eV; such large energy gaps make excitonic ISC difficult. However, the band gaps between the $S_1$ state of the guests and the $T_1$ state of the host are only 0.11–0.59 eV (Fig. 5h), which is advantageous for excitonic ISC. Therefore, the synergy action for the guest–host system is also an important factor in the phosphorescence activity of guest–host materials.

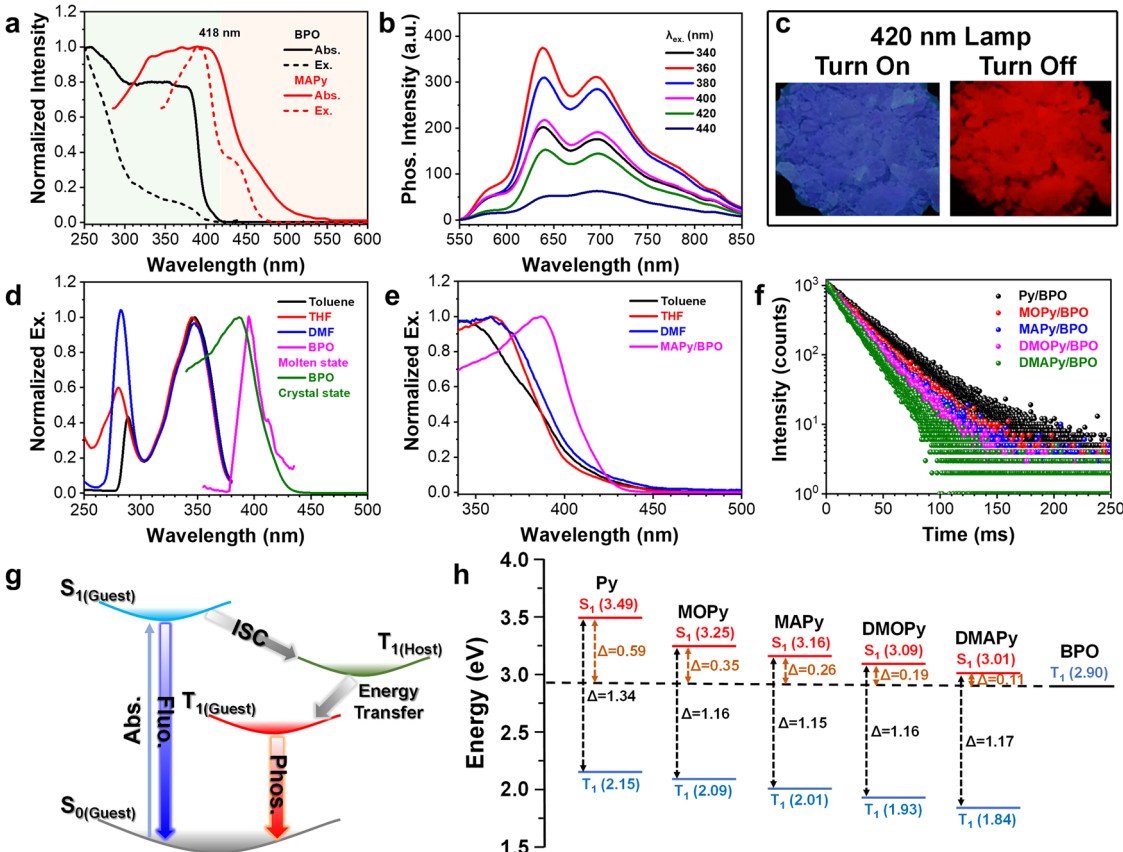

**Fig. 5 Energy transfer between guest and host. a** Excitation spectra of host **BPO** powder and guest **MAPy** powder. **b** Phosphorescence spectra of guest–host material **MAPy/BPO** at different excitation wavelengths. Excitation wavelength: 380 nm; Delayed time: 1 ms. **c** Luminescence photos of the **MAPy/BPO** powder. Excitation spectra of fluorescence (**d**)/phosphorescence (**e**) of **MAPy** in different solvent and molten state host (Concentration: $1 \times 10^{-4}$ mol/L). **f** Phosphorescence decay curves of the guests in 77 K (Ex: 380 nm; Concentration: $1 \times 10^{-4}$ mol/L; Solvent: 2-methyltetrahydrofuran). **g** Proposed transfer path between guest and host. **h** The energy levels of **BPO** and five guests.

**Application studies in time-resolved bioimaging.** Long-wavelength emissions are well-known to be beneficial in reducing tissue scattering and enhancing tissue penetration, and thus improve bioimaging quality. Encouraged by the excellent properties of these near-infrared RTP materials, an application to bioimaging was investigated. Because **DMAPy/BPO** among the materials studied exhibits the longest wavelength with a quite long lifetime, **DMAPy/BPO** and the biocompatible amphiphilic copolymer PEG-b-PPG-b-PEG (F127) were selected as nanoparticle (NP) cores and the encapsulation matrix, respectively. To ensure our **DMAPy/BPO** NPs were accessible in vivo with good RTP performance, a top-down method was employed to produce the NPs[12,47]. The phosphorescence wavelength of nanoparticles is almost the same as that in the solid-state (Supplementary Fig. 19a), but the phosphorescence lifetime is reduced to 70 ms (Supplementary Fig. 19b), and the phosphorescence quantum yield has also been reduced to 3.1%. This may be due to the small size of the nanoparticles causing the host matrix to not be able to coat the guest molecules well[12,47]. In addition, to study the morphology of the nanoparticles, the XRD curve of the nanoparticles obtained by suction filtration were tested, and the result showed that the nanoparticles have good crystallinity (Supplementary Fig. 20). To further verify the advantages of long-wavelength RTP materials, a short-wavelength ($\lambda_{Phos.} = 520$ nm) but strong intensity ($\Phi_{Phos.} = 64\%$) RTP material **DOB/BPO** reported in our previous work was selected as a control and **DOB/BPO** NPs were prepared by the same method[48]. Dynamic light scattering and transmission electron microscopy data

indicated both **DMAPy/BPO** and **DOB/BPO** NPs formed a near-spherical morphology with a mean hydrodynamic diameter of ≈ 100 nm (Fig. 6a and Supplementary Fig. 21). Both kinds of NPs revealed strong resistance to photobleaching, indicative of little change in their intensities after eight cycles of stimulation or eighty minutes of 365-nm UV light irradiation (Fig. 6b and Supplementary Figs. 22, 23). We further verified the quantitative conversion of the phosphorescence intensity with NP concentrations. The phosphorescence intensities of **DMAPy/BPO** NPs and **DOB/BPO** NPs were captured at $t = 10$ s post-excitation, and possess good linearity with the NP concentration (Fig. 6c and Supplementary Fig. 24). **DMAPy/BPO** NPs and **DOB/BPO** NPs display the main phosphorescence signals (Fig. 6d) under Dsred (575–650 nm) and GFP (515–575 nm) filters, respectively. This result is consistent with their phosphorescence spectra.

As tissue penetration is a considerable challenge for in vivo bioimaging, tissue penetration depths of the NPs were compared between **DMAPy/BPO** NPs and **DOB/BPO** NPs. The phosphorescence signals of both **DMAPy/BPO** NPs and **DOB/BPO** NPs (Fig. 6e, f) decreased with the increasing thickness of chicken breast tissue. With the advantages of RTP materials, ultra-high SBR signals were observed without covering the chicken breast tissue. However, the inherent limit with short wavelengths leads to a relatively low tissue penetration (SBR = 5.4 at thickness 7.5 mm). In contrast, the NIR phosphorescence signal of the **DMAPy/BPO** NPs can still be detected (SBR = 15) under a 12.5-mm thick coverage of chicken breast tissue. This result revealed

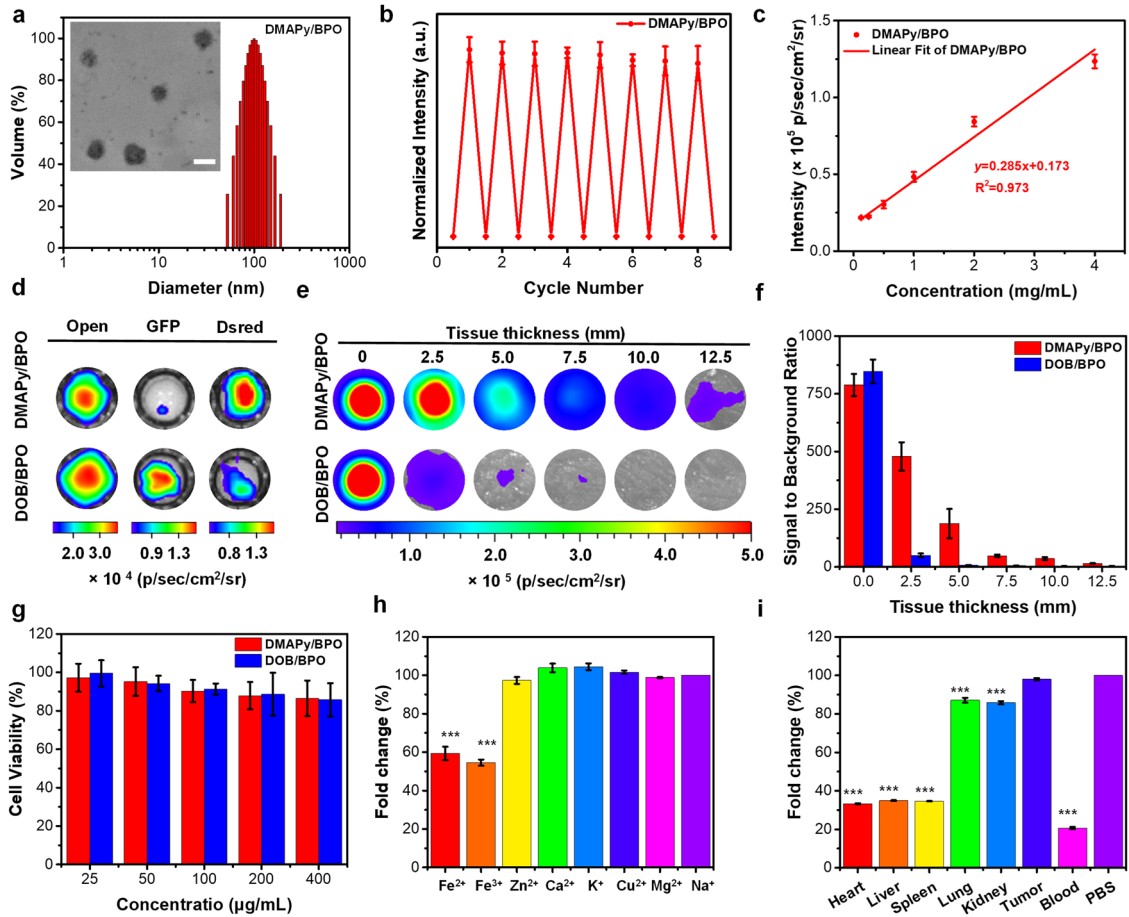

**Fig. 6 Phosphorescence properties of DMAPy/BPO and DOB/BPO nanoparticles. a** Diameter distribution of **DMAPy/BPO** nanoparticles. Inset: transmission electron microscopy image, scale bar = 100 nm. **b** The normalized phosphorescence intensities of **DMAPy/BPO** NPs as a function of cycle number of UV light irradiation ($n = 3$). **c** The phosphorescence intensities as a function of the concentration of **DMAPy/BPO** NPs ($n = 3$). **d** Phosphorescence images of **DMAPy/BPO** and **DOB/BPO** NPs (4 mg mL$^{-1}$) were captured through different filters. **e** Phosphorescence images of **DMAPy/BPO** and **DOB/BPO** NPs (10 mg mL$^{-1}$) covered with different thicknesses of chicken tissue. **f** SBR ratios from coverings with different tissue thicknesses given in (**e**). **g** Cytotoxicities of **DMAPy/BPO** and **DOB/BPO** NPs against 4T1 cells. The 4T1 cells were incubated with **DMAPy/BPO** and **DOB/BPO** NPs at different concentrations for 8 h ($n = 4$). **h** Fold change plot of phosphorescence intensities of **DMAPy/BPO** NPs in various metal ions. Error bars: mean ± standard deviation ($n = 3$). Triple asterisks represent $p < 0.01$ compared with Na$^+$. **i** Fold change plot of phosphorescence intensities for **DMAPy/BPO** NPs in different tissue homogenates. Error bars: mean ± standard deviation ($n = 3$). Triple asterisks represent $p < 0.01$ compared with PBS.

excellent deep tissue imaging from NIR emissions of NPs and moreover without excitation.

After we verified that both kinds of NPs had good cytocompatibility (Fig. 6g), we investigated the phosphorescent performance of **DMAPy/BPO** NPs in various metal ions (widespread in vivo) and tissue homogenates to confirm bioimaging feasibility in vivo. After 1 min-long irradiation using a 365-nm handheld UV lamp, the phosphorescence signals of **DMAPy/BPO** NPs incubated with different metal ions were recorded immediately under the same conditions using an IVIS bioimaging instrument. However, ferric ions (including Fe$^{2+}$ and Fe$^{3+}$) were found to quench the phosphorescence signals significantly compared with the signals in Na$^+$ (Fig. 6h). This result might be attributed to the interaction between ferric ions with outer vacant orbitals and O/N heteroatoms with lone pair electrons in **DMAPy/BPO** NPs[49] Furthermore, the **DMAPy/BPO** NPs were found to exhibit different phosphorescence quench behaviors in different tissue homogenates and blood (Fig. 6i). Compared with the phosphorescence signal in PBS, the signals of **DMAPy/BPO** NPs were significantly quenched in blood and blood-rich tissues (such as heart and liver) through the quenching

of Fe$^{2+}$ and Fe$^{3+}$ ions, which would be beneficial when imaging tumors.

Applications in intravital phosphorescence imaging were investigated. The solutions of **DMAPy/BPO** NPs and **DOB/ BPO** NPs were subcutaneously injected into Balb/c nude mice, followed by imaging with an IVIS instrument in bioluminescent mode after 1-min irradiation from the 365 nm handheld UV lamp. The images were captured 10 s after the removal of the light source. To ensure the biosafety of the UV irradiation procedure, the phosphorescence signals were activated by the handheld UV lamp at 10 mW cm$^{-2}$ power density, which is below the maximum power exposure allowed for skin irradiation (18 mW cm$^{-2}$)[47]. For comparison, fluorescence signals derived from **DMAPy/BPO** NPs and **DOB/BPO** NPs were also evaluated, simultaneously. The subcutaneous phosphorescence imaging result (Fig. 7a) in living mice reveals that both phosphorescent signals from **DMAPy/BPO** and **DOB/BPO** NPs can be observed at 10 s after excitation. The SBR of **DMAPy/BPO** NPs and **DOB/ BPO** NPs subcutaneous phosphorescence imaging at 10 s are 160 and 75 (Fig. 7b), respectively. In contrast, the fluorescence signals of **DMAPy/BPO** NPs and **DOB/BPO** NPs were hardly

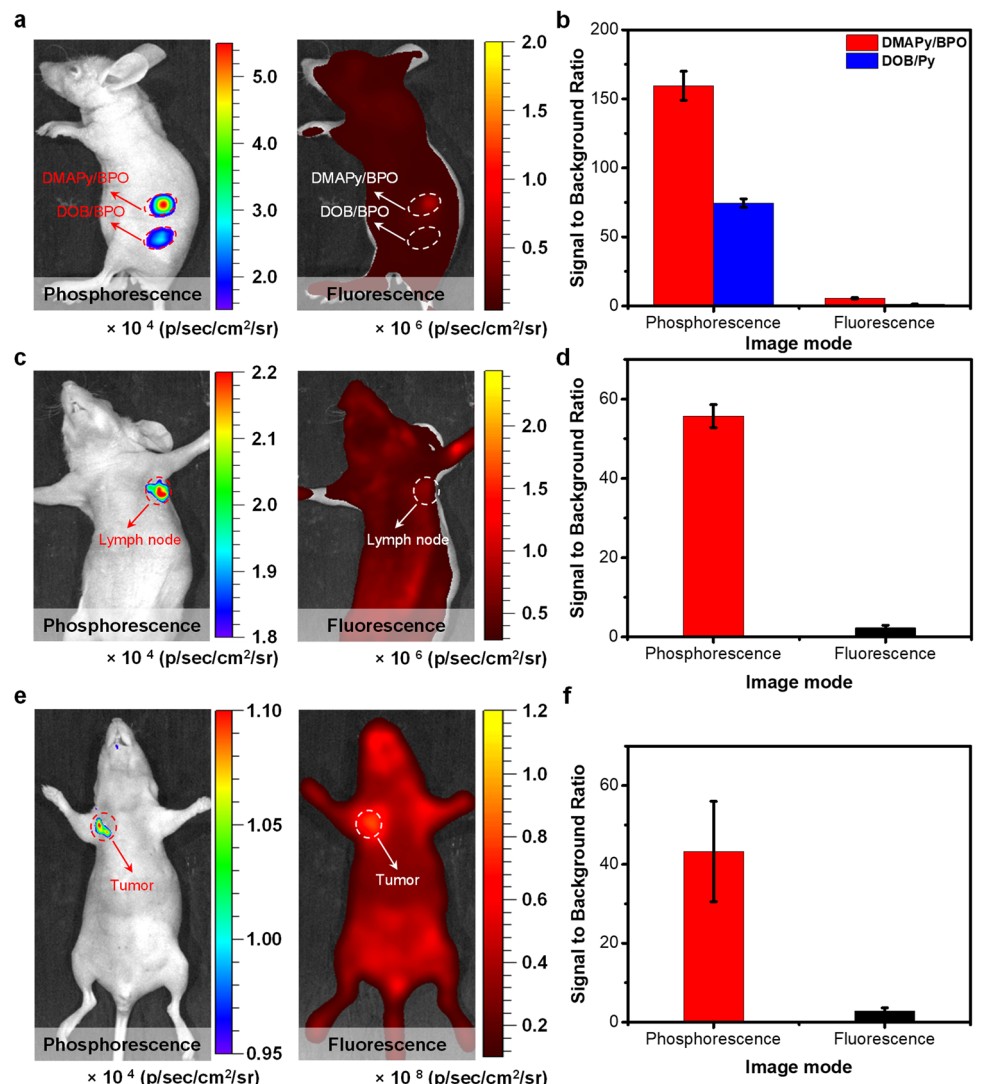

**Fig. 7 Applications in intravital phosphorescence imaging. a** Phosphorescence and fluorescence imaging of a mouse with the subcutaneous inclusions of **DMAPy/BPO** NPs and **DOB/BPO** NPs (4 mg mL$^{-1}$). Circles indicate the locations of nanoparticle injection. **b** Signal to background ratio for phosphorescence and fluorescence imaging of subcutaneous injection in live mice. **c** Phosphorescence and fluorescence imaging of lymph nodes in mice 0.5 h after intradermal injection of **DMAPy/BPO** NPs (4 mg mL$^{-1}$) into the forepaw of live mice. **d** Signal to background ratio for phosphorescence and fluorescence imaging of lymph nodes in live mice. **e** Phosphorescence and fluorescence imaging of live tumor-bearing mice 6 h after the injection of **DMAPy/BPO** NPs into the vein (4 mg mL$^{-1}$). **f** Signal to background ratio for phosphorescence and fluorescence imaging of tumor in live mice. All error bars were based on standard deviation ($n = 3$).

distinguished from the tissue autofluorescence. We note that, although the skin thickness of mice is just ~0.5 mm, short-wavelength-emitting **DOB/BPO** NPs exhibit a lower SBR than **DMAPy/BPO** NPs in subcutaneous phosphorescence imaging because of scattering from skin tissue. These results are in accordance with Fig. 5e, f and also demonstrate that long-wavelength emissions from **DMAPy/BPO** can effectively decrease tissue scattering and provide high-quality phosphorescence bioimaging. The phosphorescence imaging of lymph nodes was further investigated because lymph-node labeling is clinically important in guiding tumor surgery. The phosphorescence signal (Fig. 7c, d) of the axillary lymph node is clearly evident (SBR = 55) whereas the fluorescence signal is not distinguishable. Thus, the lymph-node imaging confirms the effectiveness of **DMAPy/BPO** NPs for phosphorescence tissue imaging.

Precise identification of complicated diseases such as cancer cells for high-performance imaging. Encouraged by the good performance of lymph-node imaging, we evaluated the

phosphorescence imaging capability in cancer diagnosis in vivo. To probe the feasibility of using long-wavelength RTP materials in such circumstances, the armpit tumor-bearing mice were prepared with 4T1 breast cancer cells. A solution of **DMAPy/BPO** NPs was injected through the tail vein into live mice. At 6 h post-injection, the signals of **DMAPy/BPO** NPs were activated by UV light for 1 min. Next, after removal of the UV lamp excitation, phosphorescence images were captured at 10 s using the IVIS instrument in the bioluminescence mode. Similarly, fluorescence imaging was recorded at the same time for comparison. The phosphorescence signal (Fig. 7e) clearly originates from the armpit tumor. Because the NIR phosphorescence emission involves no autofluorescence interference, the SBR for the phosphorescence guided armpit tumor imaging is as high as 43 (Fig. 7f).

The mice bearing armpit tumors were sacrificed, and the main tissues were excised for ex vivo phosphorescence imaging. Reticuloendothelial system organs are known to have enriched

nanomaterials. Interestingly, only the liver displayed a low phosphorescence signal; in other main organs, almost no phosphorescence signal was observed (Supplementary Fig. 25). This result might be attributed to the phosphorescence signal being quenched by the liver because of its abundant blood supply, which is in accordance with an earlier result (Fig. 6i). Furthermore, tissues with phosphorescence signals were collected and stained with H&E (Supplementary Fig. 26), verifying the presence of an armpit tumor. The main organs were stained with H&E as well. Compared with the main organs of PBS pretreatment live mice, the **DMAPy/BPO** NPs did not cause obvious damage to these organs (Supplementary Fig. 27). This work confirmed that by avoiding autofluorescence interference and reducing the tissue scattering, RTP materials (**DMAPy/BPO** NPs) with both long-wavelength and long-lifetime properties can serve as potent probes for image-guided diagnosis.

## Discussion

In conclusion, we provide a practical idea for the construction of organic RTP materials with NIR wavelength and ultralong life-time: i.e., the guests should have sufficient conjugation to reduce the $T_1$ level, and the host matrix assists the guest molecules in exciton transfer and inhibits the non-radiative transition. To the best of our knowledge, this is the first report on pure organic RTP materials with both long-wavelength (>700 nm) and ultralong lifetime (>100 ms) simultaneously. What's more, we demonstrate for the first time that pure organic NIR phosphorescence is successfully used for in vivo bioimaging. This work also represents one of the very few examples among currently available pure organic RTP materials able to be competent for in-depth in vivo bioimaging via intravenous injection in an animal model.

## Methods

**Synthesis of guest compounds**. The mixture of 1-bromopyrene or 1,6-dibro-mopyrene (10.0 mmol), boronic acid (12.0 or 24.0 mmol), Pd(PPh$_3$)$_4$ (5.0 mol%), and K$_2$CO$_3$ (5.0 mol%) were dissolved in THF (10.0 mL) and water (1.0 mL). The mixture was stirred for 12 h at 80 °C under a nitrogen atmosphere. The solvent was removed under reduced pressure and the residue were purified by column chro-matography (petroleum ether: ethyl acetate = 1:100, v:v) to afford the pure **MOPy/MAPy/DMOPy/DMAPy** compounds.

**Preparation of doped materials**. Put the corresponding amount of host and guest together, and heat the mixture to 60 °C in an air atmosphere. After the guests are completely dissolved in the molten hosts, the mixed systems are cooled to room temperature, and the mixed systems are crystallized to obtain the doped materials. The doped materials with high guest–host molar ratios (1:10, 1:100, 1:100) are using a direct weighing method, while for guest–host molar ratio (1:10000) doped materials, we use the indirect dilution method.

**Bioimaging measurement**. The amphiphilic copolymer PEG-b-PPG-b-PEG (F127) was purchased from Aladdin Ltd. Fetal bovine serum (FBS) was provided by Thermo Fisher Scientific Inc. (Waltham, MA, USA). Transmission electron microscopy (TEM) images were acquired from a JEM-2010F transmission electron microscope with an accelerating voltage of 200 kV. Dynamic light scattering (DLS) was measured on a 90 plus particle size analyzer. In vitro and in vivo phosphor-escence imaging was performed by the IVIS® Lumina II imaging system.

**Preparation of nanoparticles**. To 1 mL of the aqueous solution of F127 (10 mg), the **DMAPy/BPO** crystals (1 mg) were added. The mixture was then sonicated by a microtip-equipped probe sonicator (Branson, S-250D) for 10 min. The resultant suspension was filtered through a 0.45 μm syringe-driven filter to afford a solution of nanoparticles. And then the resultant solution was concentrated.

**In vitro phosphorescent imaging of nanoparticles solutions**. The phosphor-escent intensities of **DMAPy/BPO** and **DOB/BPO** nanoparticles were recorded using IVIS® Lumina II imaging system at $t = 10$ s after each kind of nano solutions were irradiated by 365 nm handheld UV lamp (12 W) for 1 min. The IVIS system was set in bioluminescence mode with an open/GFP/Dsred filter setting (exposure time: 17 s).

## Data availability

The authors state that the data supporting the results of this study are available in this paper and its supplementary materials. Extra data are available from the corresponding authors upon reasonable request. The data generated in this study have been deposited in the Figshare database: https://doi.org/10.6084/m9.figshare.16863643. The X-ray crystallographic coordinates for structures reported in this study have been deposited at the Cambridge Crystallographic Data Centre (CCDC), under deposition numbers 2091331, 2091334, 2091335, and 2091336. These data can be obtained free of charge from The Cambridge Crystallographic Data Centre via www.ccdc.cam.ac.uk/data_request/cif. Source data are provided with this paper.

## Code availability

No custom computer code is used in the manuscript.

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

## Acknowledgements

All animal studies were performed according to the guidelines set by the Tianjin Committee of Use and Care of Laboratory Animals, and the overall project protocols were approved by the Animal Ethics Committee of Nankai University. The accreditation number of the laboratory is SYXK(Jin) 2019-0003 promulgated by the Tianjin Science and Technology Commission. This work was supported by financial support from the National Natural Science Foundation of China (No 22071184, X.B.H; 51961160730, 51873092, and 81921004, D.D.) and the Zhejiang Provincial Natural Science Foundation of China (No LY20B020014, D.D) the National Key R&D Program of China (Intergovernmental Cooperation Project, No 2017YFE0132200, X.B.H.), and the Tianjin Science Fund for Distinguished Young Scholars (No 19JCJQJC61200, D.D.).

## Author contributions

YL, XH, and DD designed the research work and revised the manuscript. FX synthesized the materials. FX and YL carried out photophysical property measurements. HG carried out biological tissue measurements. XZ carried out density functional theory calculations. YL, XH, and DD wrote the manuscript. WD, ML, ZC, and HW edited the manuscript. All authors discussed the results and commented on the manuscript.

## Competing interests

The authors declare no competing interests.
