## [Peer Review File · Nature Communications]

Reviewers' Comments:

Reviewer #2:

Remarks to the Author:

This paper describes the preparation of organic NIR RTP materials using a guest-host system. The authors employed diaryl-substituted pyrenes as guest materials and benzophenone as a host. As the authors previously used benzophenone as a host to achieve the RTP, the novelty in this study is to use the pyrene derivatives as a guest to achieve the red-shifted phosphorescence. They achieved the phosphorescence with the maximum wavelength red-shifted up to 681 nm by extending the pi-conjugation with the electron-donating aryl groups, whereas the quantum yields are not strikingly high for these wavelength. The authors carefully elucidated the mechanism of the RTP in this guest-host system. However, the analysis methods including the MD simulation have been already employed in the previous reports and are not methodologically new, although this reviewer appreciate such careful elucidation. They further investigated the importance of benzophenone as a guest by a comparative study with analogous host molecules and discussed the crucial roles of the host material to transfer the excitons, which was, however, also already described in the previous reports by the authors. To demonstrate the utility of this long-wavelength RTP system, the authors prepared nanoparticles and nicely applied them to in vivo bioimaging, including lymph node mapping and armpit tumor labeling, which well demonstrated the advantage of this system. Overall, although the authors nicely prepared the NIR RTP materials and demonstrated their utility in in vivo imaging, the novelty of the materials used is not sufficiently high for the acceptance in this prestigious Journal. It would be suitable to publish in a more specialized Journal after addressing the following points.

- 1) As the authors claimed that the "NIR" emission > 700 nm was achieved. However, as the shortest (and highest) emission wavelength for DMAPy/BPO is 681 nm, it is reasonable to refer to this value instead of 732 nm.
- 2) Table 1 needs more detailed information including the measurement conditions. Why are there two components for lifetimes that are very close to each other?
- 3) "One-axis two-wing" strategy is not scientific wording and difficult to understand the meaning. The concept that extension of pi-conjugation with aryl substitution results in red-shifted emission is quite normal. In addition, the two roles of host have been well demonstrated previously.
- 4) No photophysical properties data for the nanoparticles are reported. Are those of nanoparticles identical to those in the solid state? How does the crystallinity in the nanoparticles affect the photophysical properties? The authors should report these data.

Reviewer #3:

Remarks to the Author:

In this manuscript, the authors developed a series of long-lifetime near-infrared room temperature phosphorescence (RTP) materials through the host-guest doped strategy. RTP materials with long wavelengths are of importance in the field of biological imaging, but as the authors have pointed out in the manuscript, although some red phosphorescence materials have been developed, there are still very few with emission wavelengths exceeding 700 nm. Additionally, the lifetimes of these red RTP materials are also quite short, which is unfavorable for practical operations in the field of biological imaging. This work used the doped strategy to obtain RTP materials with long lifetimes and long wavelengths at the same time, and successfully imaged in biological tissues, confirming the great advantages of these materials in terms of penetrability and signal to background ratio. This work has important impetus for the construction of organic RTP materials with long wavelength and long lifetime. Therefore, I recommend that the manuscript could be published on Nature Communications after the authors address the following minor issues.

1. The phosphorescence quantum yields of the Py/BPO system with different guest-host molar ratio should be provided.
2. The authors need to describe the detailed preparation conditions of the doped materials, such as temperature, air atmosphere or nitrogen atmosphere, etc.
3. What is the delay time in the delayed emission spectra (See Fig. 1c)? Please provide more detailed conditions for all graphs.
4. The environmental sensitivity of the doped materials, especially DMAPy/BPO, should be

presented, such as under the light, water or the oxygen?

5. I am interested in using MD simulations to simulate the molecular conformations of guest molecules in the host matrix. Can you further describe how to ensure the accuracy of the guest conformation simulated?

6. The authors need to supplement the phosphorescence emissions of solid or aggregated guest molecules at low temperatures in order to better discuss the morphology of the guest in the host.

7. The luminescence mechanism of the doped materials provided by the authors seems to make sense. I wonder if the energy level of the host BPO provided by the authors is gas state or solid state? If it is gas state, the authors need to recalculate the energy level of the host, because the host in the doped materials is solid state.

8. Are the nanoparticles crystalline or amorphous? It is recommended to be confirmed with XRD experiments.

9. Please provide the exactly fitting formula of phosphorescence intensity in Figure 5c.

10. The scale bar in S20 is also should be provided.

11. The statistical analysis should be described in details.

12. I wonder why the phosphorescence signals can be detected at $t=10$ s if these compounds only possess hundreds milliseconds lifetime.

Reviewer #2 (Remarks to the Author):

This paper describes the preparation of organic NIR RTP materials using a guest-host system. The authors employed diaryl-substituted pyrenes as guest materials and benzophenone as a host. As the authors previously used benzophenone as a host to achieve the RTP, the novelty in this study is to use the pyrene derivatives as a guest to achieve the red-shifted phosphorescence. They achieved the phosphorescence with the maximum wavelength red-shifted up to 681 nm by extending the pi-conjugation with the electron-donating aryl groups, whereas the quantum yields are not strikingly high for these wavelengths. The authors carefully elucidated the mechanism of the RTP in this guest-host system. However, the analysis methods including the MD simulation have been already employed in the previous reports and are not methodologically new, although this reviewer appreciate such careful elucidation. They further investigated the importance of benzophenone as a guest by a comparative study with analogous host molecules and discussed the crucial roles of the host material to transfer the excitons, which was, however, also already described in the previous reports by the authors. To demonstrate the utility of this long-wavelength RTP system, the authors prepared nanoparticles and nicely applied them to in vivo bioimaging, including lymph node mapping and armpit tumor labeling, which well demonstrated the advantage of this system. Overall, although the authors nicely prepared the NIR RTP materials and demonstrated their utility in in vivo imaging, the novelty of the materials used is not sufficiently high for the acceptance in this prestigious Journal. It would be suitable to publish in a more specialized Journal after addressing the following points.

Response: We are very grateful to the reviewer for carefully and comprehensively reviewing our manuscript, which is very helpful for us to improve the quality of the manuscript. The reviewer think that the molecular structures of the materials are not novel enough, and we sincerely accept the reviewer's criticism. But in fact, in order to construct RTP materials with both ultra-long lifetime and long wavelength simultaneously, we synthesized dozens of guest molecules over two years (**Figure R1**). Although a large number of RTP materials have been obtained, most of them are yellow-green phosphorescence emission; some of them have long wavelength, but the lifetimes are short and cannot meet the requirement of in-depth in vivo bioimaging through intravenous administration. Finally, the pyrene derivatives with high degree of conjugation were designed and synthesized as the guests. This is because high conjugation can effectively reduce the T_1 level of the molecule, thereby having a long phosphorescence wavelength. Because the benzophenone which we first used has a low T_1 level and good crystallinity (Chem. Eur. J. 2020, 26, 17376), it is an excellent host.

The low T_1 level of the host is advantageous to assist the energy transfer of the guest molecules. The good crystallinity is conducive to providing a rigid environment for the guest molecules. Not only us, but also other researchers have gradually chosen benzophenone as the host to construct RTP materials (Nat. Commun. 2021, 12, 3522; Angew. Chem. Int. Ed. 2021, 60,17138; J. Phys. Chem. Lett. 2021, 12, 4600; Adv. Opt. Mater. 2021, 9, 2100353). And in fact, we have also designed and screened a large number of new host molecules (**Figure R2**), but the doped materials with both long lifetime and ultra-long wavelength have not been obtained due to various reasons of the hosts. To obtain the optimal phosphorescence properties of doped materials (with ultra-long lifetime and long wavelength at the same time), we did not choose the new material structures and continued to choose benzophenone as the host. Actually, the novelty of this manuscript does not lie in the new structure of host molecule, but mainly the excellent phosphorescence performance and in-depth biological imaging applications. We apologize very much for not elucidating the novelty of the manuscript clearly. In the revised version, we have added a statement upon comparing with the currently reported RTP materials at the end of the Discussion part. According to the reviewer 's suggestion, in the future, we will try our best to develop doped materials with both new molecular structure and excellent phosphorescent properties.

Bioimaging is one of the most important application outlets for pure organic RTP materials, which however is just in the infancy stage. For example, intravenous injection is one of the most frequently used methods in the clinic. Nevertheless, the successful examples of RTP materials that can be used for in vivo bioimaging through intravenous injection are very limited. Furthermore, to our knowledge, there have been no previous reports on successful application of pure organic NIR phosphorescence for in vivo bioimaging. The main reason would be because of the lack of advanced RTP materials with both wavelength in the NIR region and ultra-long lifetime (>100 ms) simultaneously, as NIR wavelength is beneficial to significantly enhance tissue penetration depth and long lifetime is essential to realize afterglow imaging and obtain high signal-to-background ratio (SBR).

According to the literature, most pure organic RTP materials have emission wavelengths between 500 nm and 600 nm. This is because the low T_1 level increases the band gap between S_1 and T_1 (ΔE_{ST}), which is not conducive to intersystem crossing (ISC) of excitons, additionally, the low T_1 level easily causes excitons to be consumed non-radiatively, resulting in a significant reduction in lifetime. Therefore, assuming materials already have low T_1 , improving the ISC capability of excitons and suppressing the non-radiative transitions of excitons are the key to achieve ultra-long lifetime NIR

phosphorescence emission. **Although people have designed many materials with high conjugation, only a small part of NIR RTP materials are obtained, and these materials always face the defect of too short lifetimes (Figure R3).**

In our work, we have successfully obtained organic RTP materials with both wavelength in the NIR region and ultra-long lifetime of 102 -324 ms at the same time. These materials exhibit improved tissue penetration in bioimaging and the high-contrast tumor imaging in living mice via intravenous administration is successfully realized with a rather high SBR value of 43. This work provides a practical idea for the construction of organic RTP materials with NIR wavelength and ultra-long lifetime: The guests should have sufficient conjugation to reduce the T_1 level, and the host matrix assists the guest molecules in exciton transfer and inhibits the non-radiative transition. To the best of our knowledge, this is the first report on pure organic RTP materials with both NIR wavelength and ultra-long lifetime (> 100 ms) simultaneously. What's more, we demonstrate that pure organic NIR phosphorescence is successfully used for in vivo bioimaging. This work also represents one of the very few examples among currently available pure organic RTP materials able to be competent for in-depth in vivo bioimaging via intravenous injection in an animal model.

Finally, once again, we are grateful to the reviewer for his/her comments on our work. From the initial idea to the designing and screening of host-guest molecules to the application in vivo imaging, we have made tremendous efforts in this work. We hope this work can make some contributions to the development of organic RTP materials with NIR wavelength and long lifetime. Therefore, we sincerely hope that this work can be published with the help of reviewer, and we will continue to work hard, hoping to bring better research results to reviewers and readers.

Figure R1. Molecular structure of the guests designed by our group.

Figure R2. Molecular structure of the hosts designed and screened by our group.

Figure R3. Organic NIR RTP materials reported in previous literatures.

Q1. As the authors claimed that the “NIR” emission > 700 nm was achieved. However, As the shortest (and highest) emission wavelength for **DMAPy/BPO** is 681 nm, it is reasonable to refer to this value instead of 732 nm.

Response: Thanks to the reviewer for his/her meticulous questions. All the five doped materials have two phosphorescence peaks, which are fine structures caused by energy level vibrations. Therefore, the doped materials exhibited a wide emission band with a half-peak width of 130 nm, and there is no obvious highest peak for **MAPy/BPO**, **DMOPy/BPO**, and **DMAPy/BPO** (Figure R4a). The CIE coordinates also indicate that the doped materials have a very deep phosphorescent color, Especially for **DMAPy/BPO**, its CIE coordinates are almost beyond the range visible to the naked eye. (Figure R4b). And in fact, for **MAPy/BPO** and **DMOPy/BPO**, the intensity of the long-wavelength peak (697 nm, 713 nm) is slightly greater than the intensity of the short-wavelength peak (643 nm, 657 nm). For the rigor of the paper, we describe the wavelength range of the materials as 600 nm/657 nm – 681 nm/732 nm in the revised manuscript.

Figure R4. (a) Phosphorescence spectra of doped materials. (b) CIE coordinates of delayed emission of doped materials.

2) Table 1 needs more detailed information including the measurement conditions. Why are there two components for lifetimes that are very close to each other?

Response: Thanks again to the reviewer for his/her meticulous questions. The excitation wavelength is 380 nm and the delayed time is 1 ms.

I apologize for the confusion we caused to reviewer because we did not describe it clearly. These two lifetimes actually correspond to the two different emission peaks of the doped materials. Since these two peaks are essentially from the same triplet energy level, they are only caused by the vibration of the triplet state. Therefore, the decay rates of the two are similar (*Angew. Chem. Int. Ed.*, **2020**, 59, 10032; *Mater. Horiz.*, **2021**, DOI: 10.1039/d1mh00956g.). Another factor is caused by instrument testing. Taking **DMAPy/BPO** as an example, the two emission peaks (683 nm and 732 nm) are close to each other and have almost completely merged (Figure R4a). When we collect the photon signal at 683 nm, the photon of the long phosphorescence wavelength will also be collected. Similarly, when we collect the photon signal at 732 nm, the photon of the short phosphorescence wavelength will also be collected. Therefore, the two emission peaks have a certain mutual influence, which also causes the lifetime of the two to be closer.

We have marked the phosphorescence lifetime of the materials in the revised manuscript: a represents the lifetime of the short-wavelength phosphorescence peak, and b represents the lifetime of the long-wavelength phosphorescence peak.

3) “One-axis two-wing” strategy is not scientific wording and difficult to understand the meaning. The concept that extension of pi-conjugation with aryl substitution results in

red-shifted emission is quite normal. In addition, the two roles of host have been well demonstrated previously.

Response: We are very grateful for the professional advice of the reviewer. As the reviewer stated that establishing long lifetime NIR RTP materials inevitably means lowering the lowest triplet (T_1) level of the material, however, lower T_1 levels bring two major obstacles to phosphorescence. One of the obstacles is that lower T_1 increases the band gap between S_1 and T_1 (ΔE_{ST}), which is not conducive to intersystem crossing (ISC) of excitons. The other obstacle is that a lower T_1 level easily causes excitons to be consumed non-radiatively, resulting in a significant reduction in lifetime and intensity of phosphorescence (Figure R5). Therefore, assuming materials already have low T_1 , improving the ISC capability of excitons and suppressing the non-radiative transitions of excitons are key in achieving long-lifetime NIR phosphorescence emission. Although people have designed many materials with high conjugation, only a small part of NIR RTP materials have been obtained and these materials always face the defect of too short lifetime.

In the doped system, the host matrix can effectively solve these two problems. Therefore, we used the doping method to obtain organic phosphorescent materials with long lifetime and long wavelength for the first time. In order to better allow readers to understand our work simply and quickly, based on the reviewer's suggestion, we changed the "One-axis two-wing strategy" to the "Host-guest doped strategy" in the revised manuscript.

Figure R5. Problems in constructing ultralong near-infrared RTP materials

4) No photophysical properties data for the nanoparticles are reported. Are those of nanoparticles identical to those in the solid state? How does the crystallinity in the nanoparticles affect the photophysical properties? The authors should report these data.

Response: Thanks to the reviewer for his/her important suggestions, we have supplemented the corresponding data of nanoparticles. The nanoparticle phosphorescence spectra of **DMAPy/BPO** is almost the same as that in the solid state (Figure R6a), but the

phosphorescence lifetime is reduced to 70 ms (Figure R6b), and the phosphorescence quantum yield has also been reduced to 3.1%. Compared with solid materials, the optical properties of materials prepared into nanoparticles will decrease. Due to the performance of phosphorescence is closely related to the contact and interaction of host and guest molecules, the host molecules in nanoparticles cannot well encapsulate the guest molecules, resulting in a certain degree of decline in phosphorescence performance. We have been working hard to improve the preparation method of nanoparticles, hoping to minimize the damage of the preparation process to the phosphorescence properties of the materials. According to the reviewer's suggestion, we test the XRD curve of the nanoparticles obtained by suction filtration, and the result shows that the nanoparticles have good crystallinity (Figure R6c). This may be due to the good symmetry and small molecular volume of benzophenone which lead to its excellent crystallinity.

We have added the optical properties and XRD curve of nanoparticles to the revised manuscript.

Figure R6. (a) Phosphorescence spectra of the **DMAPy/BPO** in nanoparticles state and solid state. (b) Phosphorescence decay curves of **DMAPy/BPO** nanoparticles. (c) XRD curves of **DMAPy/BPO** nanoparticles.

Reviewer #3 (Remarks to the Author):

In this manuscript, the authors developed a series of long-lifetime near-infrared room temperature phosphorescence (RTP) materials through the host-guest doped strategy. RTP materials with long wavelengths are of importance in the field of biological imaging, but as the authors have pointed out in the manuscript, although some red phosphorescence materials have been developed, there are still very few with emission

wavelengths exceeding 700 nm. Additionally, the lifetimes of these red RTP materials are also quite short, which is unfavorable for practical operations in the field of biological imaging. This work used the doped strategy to obtain RTP materials with long lifetimes and long wavelengths at the same time, and successfully imaged in biological tissues, confirming the great advantages of these materials in terms of penetrability and signal to background ratio. This work has important impetus for the construction of organic RTP materials with long wavelength and long lifetime. Therefore, I recommend that the manuscript could be published on Nature Communications after the authors address the following minor issues.

Response: Thanks to the reviewer for his/her evaluation of our work, we will continue to work hard to bring more work to the reviewer and readers.

1. The phosphorescence quantum yields of the Py/BPO system with different guest-host molar ratio should be provided.

Response: Thanks to the reviewer's suggestion. We have tested the phosphorescence quantum yields of the **Py/BPO** doped materials. And the phosphorescence quantum yields of 1:50, 1:100, 1:500, 1:1000, 1:5000, 1:10000, and 1:50000 are 3.3%, 5.5%, 7.8%, 9.2%, 6.8%, 5.1% and 3.6%, respectively. We have added the test results in the revised manuscript.

2. The authors need to describe the detailed preparation conditions of the doped materials, such as temperature, air atmosphere or nitrogen atmosphere, etc.

Response: Thanks to the reviewer's suggestion. Put the corresponding amount of host and guest together, and heat the mixture to 60 °C in air atmosphere. After the guests are completely dissolved in the molten hosts, the mixed systems are cooled to room temperature, and the mixed systems are crystallized to obtain the doped materials. The doped materials with high guest-host molar ratio (1:10, 1:100, 1:1000) are using direct weighing method, while for guest-host molar ratio (1:10000) doped materials, we use the indirect dilution method. We have added the detailed preparation method in the revised Supporting Information.

3. What is the delay time in the delayed emission spectra (See Fig. 1c)? Please provide more detailed conditions for all graphs.

Response: Thanks to the reviewer's suggestion. The excitation wavelength is 380 nm and the delayed time is 1 ms. We have added the test conditions in the revised manuscript.

4. The environmental sensitivity of the doped materials, especially **DMApy/BPO**, should be presented, such as under the light, water or the oxygen?

Response: Thanks to the reviewer's suggestion. The doped materials we prepared are kept in the air for up to one year, and their phosphorescence intensity is almost unchanged, so they are very stable to oxygen. Since the afterglow effect of **DMApy/BPO** is not as obvious as that of **MAPy/BPO**, in order to more intuitively reflect the stability of the materials, we soak the doped materials **Py/BPO** and **MAPy/BPO** in water for 24 hours, and they still have a strong red afterglow. Therefore, it is relatively stable to water (Figure R7). The materials also show strong light stability. They can still maintain excellent red afterglow after being continuously irradiated for 24 hours (365 nm, 40 $\mu\text{W}/\text{cm}^2$) (Figure R7). We have added the test results in the revised manuscript.

Figure R7. (a) Phosphorescence photographs of **Py/BPO** in different state. (b) Phosphorescence photographs of **MAPy/BPO** in different state.

5. I am interested in using MD simulations to simulate the molecular conformations of guest molecules in the host matrix. Can you further describe how to ensure the accuracy of the guest conformation simulated?

Response: We are very grateful to the reviewer for his/her interest in our calculation method (Chem. Sci. 2021, 12, 6518). The molecular packing of host matrix and guest-doped matrix are similar because of their almost the same spectra of XRD curves (Figure R8a). According to the X-ray curve in experiment, we setup the guest-doped matrix model based on the original **BPO** crystal structure and simply replace two neighbor host small **BPO** molecules by single large **Py** molecule to keep the molecular packing of host

matrix. In addition, the simulated guest conformation is quite stable, supported by the small fluctuation of the RMSD value of the backbone of guest molecules as a function of simulation time (Figure R8b). Thus, the configuration of **Py** molecule is accurate.

Figure R8. (a) The XRD curves spectra of **BPO** sample and **Py/BPO** doped material. (b) The RMSD analysis of **Py** in the doping system.

6. The authors need to supplement the phosphorescence emissions of solid or aggregated guest molecules at low temperatures in order to better discuss the morphology of the guest in the host.

Response: Thanks to the reviewer's suggestion. According to the reviewer's suggestion, we have supplemented the phosphorescence spectra of five guests at low temperature (77 K). But the results show that even at low temperature, the solid guests have no obvious phosphorescence emission in the long wavelength region (Figure R9a). The phosphorescence spectra of the doped materials highly coincide with the phosphorescence spectra of the solution guests (Figure R9b). Therefore, we think that the guests do not exist in the host matrix in a solid state. Although the focus of this work is to improve the phosphorescence properties of doped materials and perform in vivo imaging, we are still very grateful for the research idea provided by the reviewer, and we will focus on the morphology of doped materials in our future work.

Figure R9. (a) Phosphorescence spectra of the solid guests in 77 K (Ex: 380 nm; Delayed time: 1 ms) (b) Phosphorescence spectra of the solution guests in 77 K (Delayed time: 1 ms; Ex: 380 nm; Concentration: 1×10^{-5} mol/L; Solvent: 2-methyltetrahydrofuran).

7. The luminescence mechanism of the doped materials provided by the authors seems to make sense. I wonder if the energy level of the host BPO provided by the authors is gas state or solid state? If it is gas state, the authors need to recalculate the energy level of the host, because the host in the doped materials is solid state.

Response: Thanks to the reviewer's suggestion. Before calculating, we also considered the question raised by the reviewer. Therefore, we calculated the energy level of the host in the solid state. In order to allow readers and reviewer to understand the paper more easily, we have added notes to the revised manuscript ("The energy level of the solid host is calculated.").

8. Are the nanoparticles crystalline or amorphous? It is recommended to be confirmed with XRD experiments.

Response: Thanks to the reviewer's suggestion. We tested the XRD curve of the nanoparticles obtained by suction filtration, and the result shows that the nanoparticles have good crystallinity (Figure R6c).

Figure R6 (c) XRD curves of **DMAPy/BPO** nanoparticles.

9. Please provide the exactly fitting formula of phosphorescence intensity in Figure 5c.

Response: Thanks to the reviewer's suggestion. We have added the fitting formula in Figure 5c of the revised manuscript.

10. The scale bar in S20 is also should be provided.

Response: Thanks to the reviewer's suggestion. We have added the scale bar in Figure S20 and renewed the description in Figure S20 legend in the revised supporting information.

Figure R10. The phosphorescence signals of different organs (1-heart, 2-liver, 3-spleen, 4-lung and 5-kidney) from tumor-bearing mice at 6 h post intravenous injection of **DMAPy/BPO** nanoparticles. scale bar=1 cm.

11. The statistical analysis should be described in details.

Response: Thanks for the reviewer for pointing out this issue. We changed this description in the Supporting Information, which is listed as follows:

The statistical analysis of normal hypothesis for the phosphorescence intensity of ions or tissues solution in each group was conducted using SPSS 21.0 statistical software (Lead Technologies, Chicago, USA). The phosphorescence intensity of ions or tissues solution in each group were expressed as mean \pm standard deviation (SD). The differences between the groups were analyzed using a one-way analysis of variance (ANOVA), followed by Tukey's multiple comparisons test. Significant differences were defined as $p < 0.05$. (**).

12. I wonder why the phosphorescence signals can be detected at $t = 10$ s if these compounds only possess hundred milliseconds lifetime.

Nankai University

State Key Laboratory of Medicinal Chemical Biology, College of Life Sciences

Response: We thank the reviewer for the careful consideration. Actually, this is also a question that once bothered us. After our discussion, we concluded that phosphorescence lifetime is defined as the time when the phosphorescence intensity decays to the original $1/e$. The quantification of its intensities decreased as an exponential curve. However, the phosphorescence signals of RTP materials are still emitting photons after its lifetime. Although these photons are relatively weak and rare, they can still show obvious phosphorescence signals when these photons are enriched. Thus, these phosphorescence signals always much longer the lifetime of RTP materials and can also be captured by the detector or even our naked eyes. In this work, the detector of IVIS System with high sensitivity was selected for in vivo imaging of our RTP materials. The photons of phosphorescence signals can be enriched and amplified by the IVIS system within the exposure time. Therefore, the phosphorescence signals can be detected at $t=10$ s and be applied in bioimaging. Further, similar experiments were reported in previous literatures. (Adv. Mater. 2017, 29, 1606665; Adv. Mater. 2020, 32, 2006752; Small 2021, 17, 2005449; Adv. Mater. 2021, 33, 2007811).

Finally, we very thank the reviewer very much for the insightful suggestions to improve our manuscript!

Reviewers' Comments:

Reviewer #2:

Remarks to the Author:

The revised manuscript sufficiently addressed all of my concerns and the revision made the important advances of the present results clearer. I am convinced by the authors' explanation and revisions and would like to support the publication of this manuscript as this form.

Reviewer #3:

Remarks to the Author:

This work is useful and interesting.

REVIEWERS' COMMENTS

Reviewer #2 (Remarks to the Author):

The revised manuscript sufficiently addressed all of my concerns and the revision made the important advances of the present results clearer. I am convinced by the authors' explanation and revisions and would like to support the publication of this manuscript as this form.

Response: Thanks to the reviewer for his/her support, we will continue to work hard to strive for better work for reviewers and readers.

Reviewer #3 (Remarks to the Author):

This work is useful and interesting.

Response: Thank the reviewer for his/her affirmation, we will continue to work hard.